# Research

materials science

carbon steel, $CO_2$ corrosion, graphene, coating

**Author for correspondence:**
Haijun Hu
e-mail: huhaijun@mail.xjtu.edu.cn

# Corrosion resistance of graphene/waterborne epoxy composite coatings in $CO_2$-satarated NaCl solution

Hao Xu[1], Haijun Hu[1], Hongmei Wang[2], Yongjun Li[2] and Yun Li[1]

[1]Department of Process Equipment and Control Engineering, Xi'an Jiaotong University, Xi'an 710049, People's Republic of China
[2]Gas Production Technology Research Institute, NO.1 Gas Production Plant of Changqing Oilfield Company, Jingbian 718500, People's Republic of China

(iD) HX, 0000-0002-9277-1886

This study investigated the corrosion resistance of graphene/waterborne epoxy composite coatings in $CO_2$-satarated NaCl solution. The coatings were prepared by dispersing graphene in waterborne epoxy with the addition of carboxymethylcellulose sodium. The structure and composition of the coatings were characterized by scanning electron microscopy, transmission electron microscopy, Fourier-transform infrared and Raman spectroscopies. The corrosion resistance of the composite coatings was investigated by potentiodynamic polarization measurements and electrochemical impedance spectroscopy. Composite coatings with more uniform surfaces and far fewer defects than blank waterborne epoxy coatings were obtained on 1020 steel. The 0.5 wt% graphene/waterborne epoxy composite coating exhibited a much lower corrosion rate and provided better water resistance properties and long-term protection than those of the blank epoxy coating in $CO_2$-satarated NaCl solution.

# 1. Introduction

The $CO_2$ corrosion of pipelines is attracting more and more attention in the oil and gas industry because of the $CO_2$-enhanced oil recovery [1]. In this process, by injecting $CO_2$ into oil and gas fields, the residual oil and gas can be pushed out to improve oil recovery. $CO_2$ corrosion is the main form of internal corrosion failure in oil and natural gas pipelines [2]. During the $CO_2$ flooding test in the Little Creek Oilfield, USA, the wall of the oil extraction pipe was corroded and perforated within five months with a corrosion rate of 12.7 mm yr$^{-1}$ [3]. Pipeline corrosion resulted in billions of dollars in losses to the oil and

**Table 1.** Chemical composition of the 1020 steel used for the experiments (wt%).

| C | Si | Mn | P | S | Ni | Cr | Cu | Fe |
|---|----|----|----|----|----|----|----|----|
| 0.17–0.23 | 0.17–0.37 | 0.35–0.65 | ≤0.035 | ≤0.035 | ≤0.30 | ≤0.25 | ≤0.25 | balance |

gas industry. Numerous reports have shown that the rupture of oil and gas pipelines caused by corrosion has resulted in oil spillages causing environmental pollution. Further ecological damage and economic losses are then caused by dealing with such environmental pollution [4,5].

These losses associated with $CO_2$ have promoted studies on corrosion prevention methods in the oil and gas industry. Coatings [6], inhibitors [7,8] and adding alloying elements [9,10] are the most common strategies to minimize corrosion of the substrate metal. In particular, organic coatings have been considered as an effective way to retard corrosion owing to their excellent properties and stability.

Traditional organic coatings are mainly solvent based. The release of volatile organic compounds during the coating and service processes can be harmful to the environment and the health of construction workers. Unfortunately, the protective effects of waterborne coatings, such as their film-forming properties, water resistance properties and shielding properties, are significantly lower than those of solvent-based coatings. We therefore considered adding graphene to improve the corrosion resistance of waterborne epoxy coatings for application in oil and gas industry environments.

Graphene is a flat film carbon material that is composed of monolayer $sp^2$-hybridized carbon atoms. Graphene has high thermal and chemical stability and can isolate the substrate from the corrosive medium to provide effective corrosion inhibition [11]. Chen *et al.* [12] first studied the antioxidant capacity of graphene grown on the surface of pure Cu and a Cu/Ni alloy through chemical vapour deposition (CVD). It has been reported that graphene obtained through CVD can tremendously improve the resistance of the base metal to corrosion in a short time [13–16], however, CVD cannot yield defect-free graphene films. Over the long term, $O_2$ and $H_2O$ molecules will penetrate through the defects and corrode the metal substrate [17].

Many reports have indicated that graphene composite coatings can provide considerable corrosion inhibition. The addition of graphene and functionalized graphene in organic coatings can further improve the corrosion performance in marine environments [18–20]. Ye *et al.* [21] intercalated the silanized trianiline precursor into graphene sheets to prepare the aniline trimer (AT)-functionalized graphene (SAT-G). The silanized AT could disperse graphene sheets in the epoxy resin more effectively. The composite epoxy matrix could provide much better corrosion resistance over the long term by adding the appropriate amount of SAT-G (0.5 wt%). Yang *et al.* [22] synthesized the 3,4,9,10-perylene tetracarboxylic acid-graphene (PTCA-G) composite and investigated the anti-corrosion properties of composite coatings containing PTCA-G. The composite coating showed excellent corrosion resistance when containing the PTCA-G composite with a 10 : 4 volume ratio of PTCA and G.

Although graphene-reinforced epoxy coatings have shown improved corrosion resistance in many applications, there is still a lack of relevant research in oil and gas production environments where aggressive $CO_2$ and sometimes high concentrations of $Cl^-$ are both present. In this study, we investigated the corrosion resistance of graphene/waterborne epoxy composite coatings in $CO_2$-satarated NaCl solution and discussed the corrosion mechanism of the composite coatings.

## 2. Experimental

### 2.1. Material and methods

Around 1020 steel specimens were cut into 10 mm × 10 mm × 2 mm and then sealed in epoxy resin with an exposure area of 1 cm² (10 mm × 10 mm) for electrochemical measurements. The chemical composition of 1020 steel used in this study is shown in table 1. Prior to coating, the test surfaces of the specimens were abraded using sandpaper with grit grades of 280, 400, 600, 800, 1200 and 1500. The surfaces were then polished with a flannel polishing cloth (≤1 µm), degreased with ethanol and washed with deionized water. The prepared specimens were stored in a desiccator before testing.

F0704 two-component waterborne epoxy was purchased from Dongguan Rongzhong Chemical Co., Ltd. Graphene powder (purity: 99.7%, 4–7 nm/6 × 6 µm) was purchased from Beijing Forsman Technology Co., Ltd. Diamino-functionalized polyethylene glycol (NH2-PEG-NH2, purity ≥ 95%) was purchased from Dongguan Aoda Co., Ltd. Carboxymethylcellulose sodium (CMC)

(viscosity:1500–3100 mpa.s, USP grade), fumed silica (purity: 99.8%) and other reagents were supplied from Shanghai Macklin Biochemical Co., Ltd.

## 2.2. Electrolyte composition

According to the composition data collected from a gas production plant site, the $Cl^-$ concentration in formation water can be as high as $60\,000$–$100\,000\,mg\,l^{-1}$. Therefore, electrochemical experiments were conducted in high chloride environment aqueous solutions with 10 wt% NaCl at 25°C and atmospheric pressure. The electrolyte solution was purged with $CO_2$ for 8 h to deoxygenate and saturate with $CO_2$ before the specimen was inserted into the solution. $CO_2$ was continuously introduced into the electrolyte solution during the electrochemical experiment.

## 2.3. Preparation of graphene/waterborne epoxy composite coating

The composite coatings were prepared by the dispersion of different mass fractions of graphene in waterborne epoxy matrix. CMC was used as the dispersant and fumed silica was used as the anti-sediment agent. Different weight ratios (0%, 0.25% and 0.50%) of graphene and a small amount of diamino-functionalized polyethylene glycol were added into the waterborne epoxy and the composite was then dispersed by ultrasonication for 0.5 h. Epoxy curing agent was added with a 2 : 1 weight ratio of epoxy matrix and curing agent. Then, the composite was dispersed by ultrasonication for another 0.5 h and uniformly coated on the test surface of 1020 steel using a wire bar coater (20 μm). The sample was then cured at 25°C for 24 h.

## 2.4. Characterization of coatings

An infrared spectrometer (Nicolet iS50) was used to record the Fourier-transform infrared (FTIR) spectra. Raman spectra were collected on a laser Raman spectrometer (LabRAM HR Evolution). Scanning electron microscopy (SEM) images were recorded on using an MAIA3 LMH microscope. In order to characterize the microscopic morphology of the graphene and composite coatings, an FEI Tecnai G2 F20 microscope was employed to collect the transmission electron microscopy (TEM) images.

## 2.5. Electrochemical measurements

The anti-corrosion properties of different coatings in $CO_2$-satarated NaCl solution were studied by measurements of polarization curve and electrochemical impedance spectroscopy (EIS). A glass cell equipped with a three-electrode system was employed in this study. The working electrode was a bare or coated steel 1020 specimen. A platinum electrode was employed as the counter electrode and a saturated calomel electrode was used as the reference electrode. The polarization curves were collected from $-0.45\,V$ to $0.45\,V$ versus open circuit potential at the scan rate of $0.5\,mV\,s^{-1}$ and fitted by CVIEW 2.6. The EIS data were recorded from 100 kHz to 0.01 Hz with a 40 mV amplitude signal and fitted by ZSIMPWIN 3.30d. Electrochemical workstation (CS350) was used to carry out all electrochemical tests.

# 3. Results and discussion

## 3.1. Characterization of composite coatings

Raman spectra were recorded to identify if interactions between the CMC and graphene sheets occurred during coating preparation. In figure 1a, the pristine graphene exhibited two characteristic peaks corresponding to the G band at $1583\,cm^{-1}$ and the D band at $1345\,cm^{-1}$, respectively [23,24]. The G band corresponds to the structure of $sp^2$-bonding carbon atoms, while the D-band is owing to the breaking mode near the K zone boundary [25].

Graphene dispersed by CMC is referred to as CMC–graphene in the rest of this paper. The Raman spectrum of CMC–graphene is shown in figure 1b. The G band for graphene ($1583\,cm^{-1}$) was blue-shifted to $1586\,cm^{-1}$ for CMC–graphene, indicating strong interaction between the CMC and graphene sheets. The integral areas value of the D band to G band ($I_D/I_G$) was used to evaluate the quality of graphene sheets. The $I_D/I_G$ ratio of pristine graphene was 1.12 which increased to 1.25 for

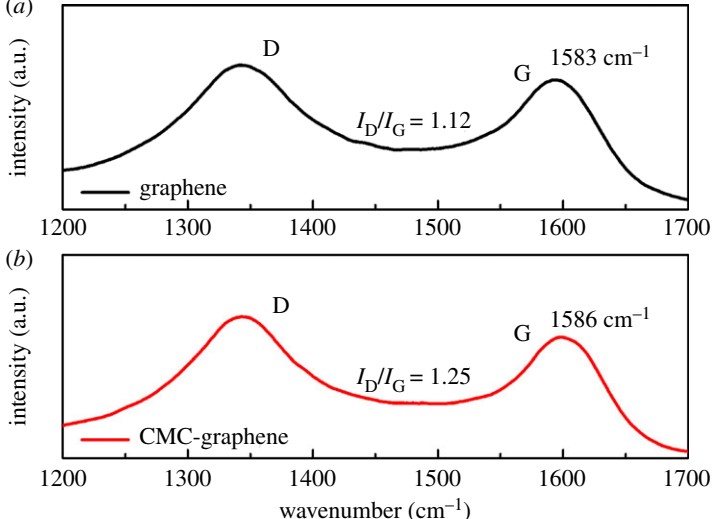

**Figure 1.** Raman spectra of (*a*) pristine graphene sheets and (*b*) CMC–graphene sheets.

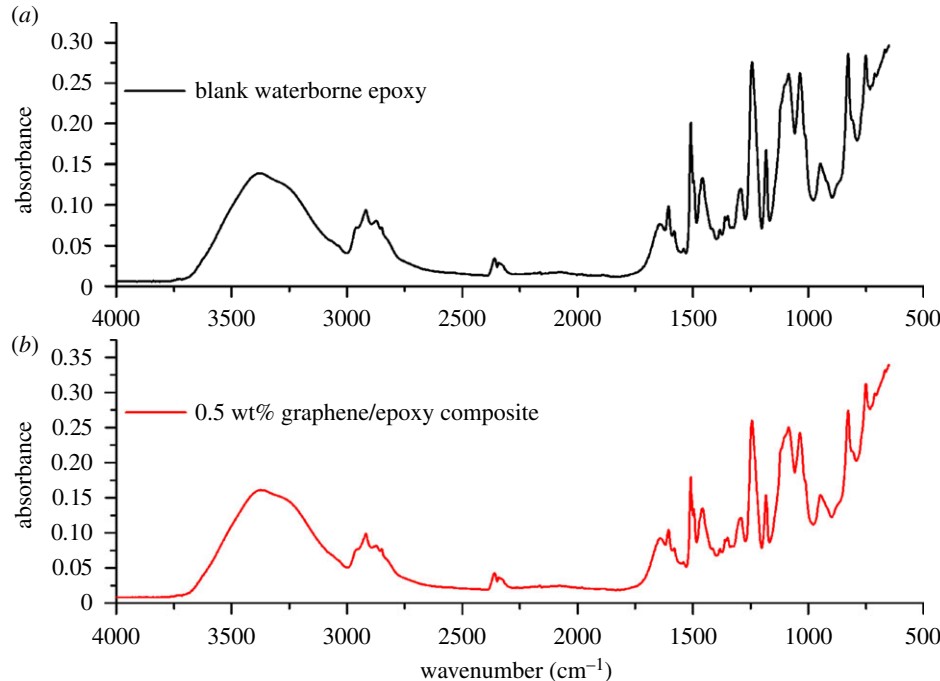

**Figure 2.** FTIR spectra of (*a*) blank waterborne epoxy coating and (*b*) 0.5 wt% graphene/waterborne epoxy composite coating.

CMC–graphene. This suggested that the degree of disorder for the CMC–graphene sheets was higher than that for the pristine graphene.

FTIR spectroscopy was also used to analyse the interaction between the graphene sheets and epoxy matrix. Figure 2*a* shows the FTIR spectrum of the blank waterborne epoxy coating. Absorption peaks at 1606 cm$^{-1}$ and 1508 cm$^{-1}$ are owing to stretching vibrations of the phenyl ring. The peak near 2800–3000 cm$^{-1}$ is owing to the C–H vibration of the aliphatic group. The peak in 1244 cm$^{-1}$ and the broad absorption at 3382 cm$^{-1}$ are assigned to C–O–C and –OH respectively. Absorption at 916 cm$^{-1}$ and 827 cm$^{-1}$ are assigned to epoxy groups. Figure 2*b* shows the FTIR spectrum of the 0.5 wt% graphene/waterborne epoxy composite coating. Introducing graphene did not evidently result in new functional groups. This indicated that graphene was physically dispersed in the epoxy matrix without chemical modification.

The microscopic morphology of pristine graphene was investigated by SEM, as shown in figure 3*a*. The flake structure of pristine graphene was clearly observed. Agglomeration and stacking were also observed, which may result in degradation of the properties of graphene. The dispersion of graphene was investigated by TEM. Figure 3*b* and *c* shows TEM images of the dispersion of untreated pristine

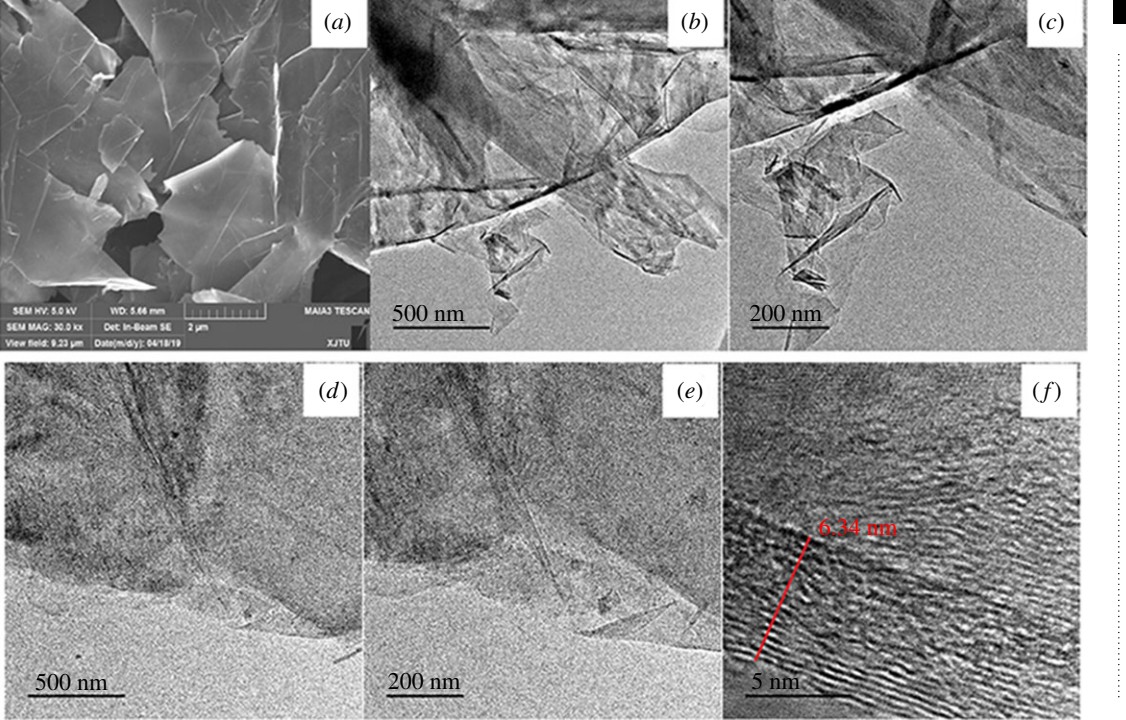

**Figure 3.** (*a*) SEM image of pristine graphene; (*b,c*) TEM images of pristine graphene; (*d,e,f*) TEM images of graphene dispersed by CMC.

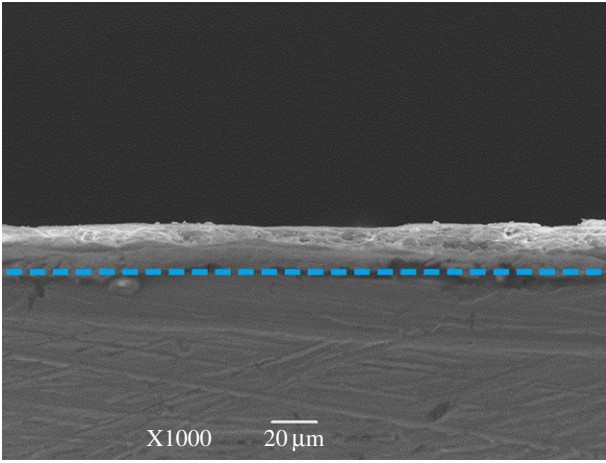

**Figure 4.** The cross-sectional SEM image of 0.5 wt% graphene/waterborne epoxy composite coating on the steel 1020 test surface.

graphene. The untreated graphene sheets were highly agglomerated and showed many wrinkles. TEM images of graphene dispersed by CMC are shown in figure 3*d*−*f*. A thin film was observed with no obvious aggregation and far less wrinkled surfaces. The high-resolution TEM image in figure 3*f* shows the average thickness of the CMC–graphene is about 6–7 nm. This indicated that monolayer or few-layer graphene sheets were obtained by CMC dispersion.

Based on the above characterization, waterborne epoxy coatings with graphene dispersed by CMC were expected to exhibit better corrosion resistance performance than those with pristine graphene. Therefore, composite coatings mentioned in the following discussion refer to waterborne epoxy coatings containing CMC–graphene. The graphene/waterborne epoxy composite was coated on the test surface of the steel 1020 using a wire bar coater. The thickness of the coating was investigated by SEM. The SEM image in figure 4 suggested that a uniform graphene/waterborne epoxy composite coating with a thickness of $20 \pm 2 \, \mu m$ was successfully obtained on the test surface of the working electrode.

Figure 5 shows SEM images of the surface topography of the blank epoxy coating and 0.5 wt% graphene/waterborne epoxy composite coating. The SEM image of the blank waterborne epoxy coating in figure 5*a* shows a surface topography containing many pores and inhomogeneous defects that is owing to the rapid water evaporation during the curing process of waterborne epoxy coatings.

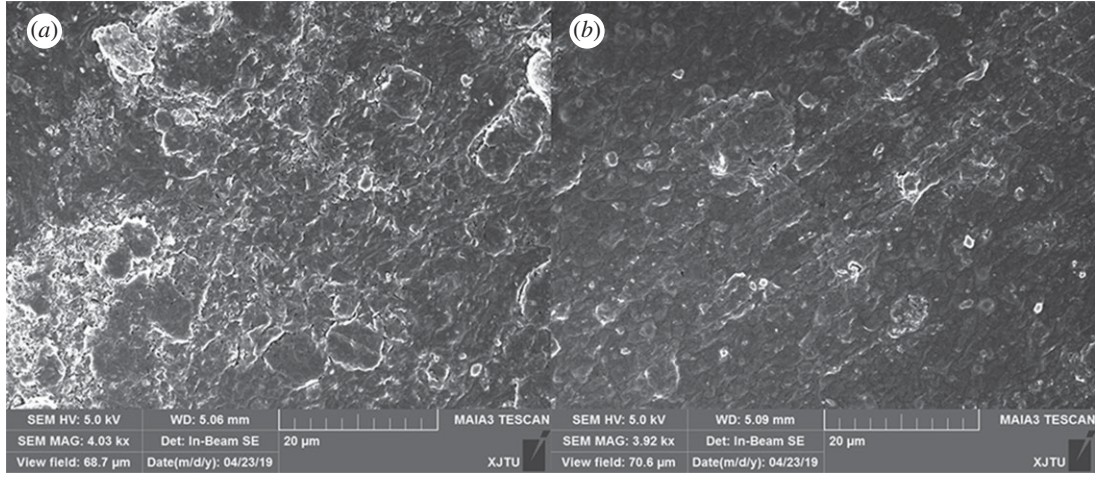

**Figure 5.** SEM images of the surface topography of (*a*) blank epoxy coating and (*b*) 0.5 wt% graphene/waterborne epoxy composite coating.

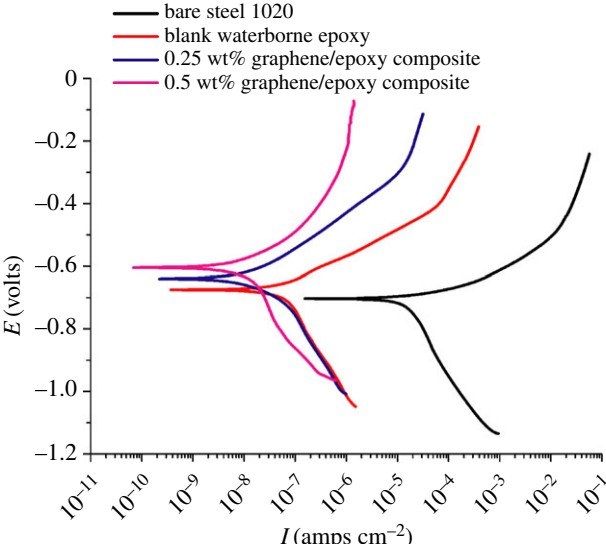

**Figure 6.** Polarization curves of the bare, blank waterborne epoxy, 0.25 wt% graphene/epoxy composite and 0.5 wt% graphene/ epoxy composite-coated electrodes in $CO_2$-satarated 10 wt% NaCl solution after 2 h of immersion.

These pores and defects will serve as active channels for aggressive species, and it is not surprising that the corrosion resistance of waterborne epoxy coatings are significantly lower than those of solvent-based epoxy coatings. The SEM image of the waterborne epoxy coating incorporating graphene in figure 5*b* showed a more uniform and smooth surface with far fewer cracks. This is because the graphene was well dispersed in the epoxy matrix, which blocked the diffusion pathways and slowed the evaporation of water [26]. The graphene/waterborne epoxy composite coating could therefore retard the permeation of corrosive medium and potentially improve the corrosion resistance performance.

## 3.2. Polarization curve measurements

The polarization curves of bare steel 1020 and steel 1020 coated with blank waterborne epoxy, 0.25 wt% and 0.5 wt% graphene/waterborne epoxy composite after immersion in $CO_2$-satarated 10 wt% NaCl solution for 2 h at 25°C are shown in figure 6. The parameters obtained from the Tafel extrapolation method are shown in table 2, in which $I_{corr}$ and $E_{corr}$ stand for the corrosion current density and the corrosion potential respectively. $B_a$ and $B_c$ represent the slope of the anodic and cathodic polarization, respectively. $\eta$ stands for the inhibition efficiency calculated by equation (3.1):

$$\eta = 1 - \frac{I'_{corr}}{I^0_{corr}}, \tag{3.1}$$

**Table 2.** Parameters obtained from polarization curves in figure 6.

| | bare steel 1020 | blank waterborne epoxy | 0.25%wt graphene/epoxy composite | 0.5%wt graphene/ epoxy composite |
|---|---|---|---|---|
| corrosion rate (mm a$^{-1}$) | 0.381 | $7.93 \times 10^{-4}$ | $1.64 \times 10^{-4}$ | $4.99 \times 10^{-5}$ |
| $I_{corr}$ (amps cm$^{-2}$) | $3.73 \times 10^{-5}$ | $6.74 \times 10^{-8}$ | $1.39 \times 10^{-8}$ | $4.24 \times 10^{-9}$ |
| $E_{corr}$ (volts) | $-0.705$ | $-0.676$ | $-0.639$ | $-0.604$ |
| $B_a$ (mV dec$^{-1}$) | 41.3 | 84.7 | 128 | 167 |
| $B_c$ (mV dec$^{-1}$) | $-357$ | $-329$ | $-289$ | $-345$ |
| $\eta$ | — | 99.79% | 99.96% | 99.98% |

where $I_{corr}^0$ stands for the corrosion current density of the uncoated steel 1020. $I'_{corr}$ represents the corrosion current density of the coated electrode [27]. The $E_{corr}$ of the 0.5 wt% graphene/epoxy composite coating ($-0.604$ V) was more positive than that of the blank waterborne epoxy coating ($-0.676$ V), which showed that adding graphene lowered the corrosion tendency. The anodic and cathodic polarization current densities of the 0.25 wt% graphene/epoxy composite coating were around 2–3 orders of magnitude lower than that of bare 1020 steel with $\eta = 99.96\%$. When graphene was added with a mass fraction of 0.5%, the corrosion rate was further reduced compared with that of the 0.25 wt% graphene/epoxy composite coating with $\eta = 99.98\%$. Table 1 shows that $B_a$ increased with the addition of graphene while $B_c$ did not change significantly. The increase of $B_a$ indicates the transition of anodic reaction. With the addition of graphene, the intensity of the Fe element of graphene/epoxy composite coating at the interface of the substrate metal and the composite coating is stronger than that of the pure epoxy coating, and the increase in the intensity of Fe element indicates the deposition of iron ions at the interface [28]. Therefore, the graphene distributed inside the composite coating was thought to impede the diffusion of Fe ions which retarded the anodic reaction.

## 3.3. Electrochemical impedance spectroscopy measurements

EIS measurements were used to study the resistance and porosity of the coatings. Figure 7 shows Bode plots of the blank waterborne epoxy, 0.25 wt% and 0.5 wt% graphene/waterborne epoxy composite coatings after an initial period of immersion of 0−90 min. All coatings exhibit only one time constant in the high-frequency range, which corresponds to the capacitance of the coating within 30 min of immersion time. In this period, the coating is equivalent to an insulating layer with a large resistance and small capacitance, which isolates moisture from the base metal [29].

With the increase of immersion time, the second time constant associated with the capacitance of a double layer appears in the low-frequency range, as shown in figure 7a. It shows that the corrosion medium has penetrated into the interface and forms a double layer between the blank waterborne epoxy coating and the substrate metal after 90 min of immersion time.

In comparison, the appearance of the second time constant in the low-frequency range for the 0.25 wt% graphene/waterborne epoxy composite coating was delayed, while the second time constant for the 0.5 wt% graphene composite coating hardly appeared. This indicates that the permeation of the corrosive medium was inhibited.

Figure 8 shows Nyquist plots of the blank waterborne epoxy coating and composite coatings during 96 h of immersion time. The EIS profiles for the composite coatings at various immersion times showed two capacitive loops. For the blank waterborne epoxy coating in figure 8a, the radii of the two capacitive loops decreased dramatically with increasing immersion time and then stabilized after 72 h. This indicated poor water isolating properties of the blank epoxy coating.

For the 0.25 wt% graphene/waterborne epoxy composite coating in figure 8b, the radii of the two capacitive loops were larger than those of the blank waterborne epoxy coating. However, the impedance modulus still decreased significantly during the 96 h of immersion.

The impedance modulus of the 0.5%wt graphene/waterborne epoxy composite coating in figure 8c was around two orders of magnitude higher than that of the blank waterborne epoxy coating. The radii of the two capacitive loops remained relatively constant during the overall immersion time. This

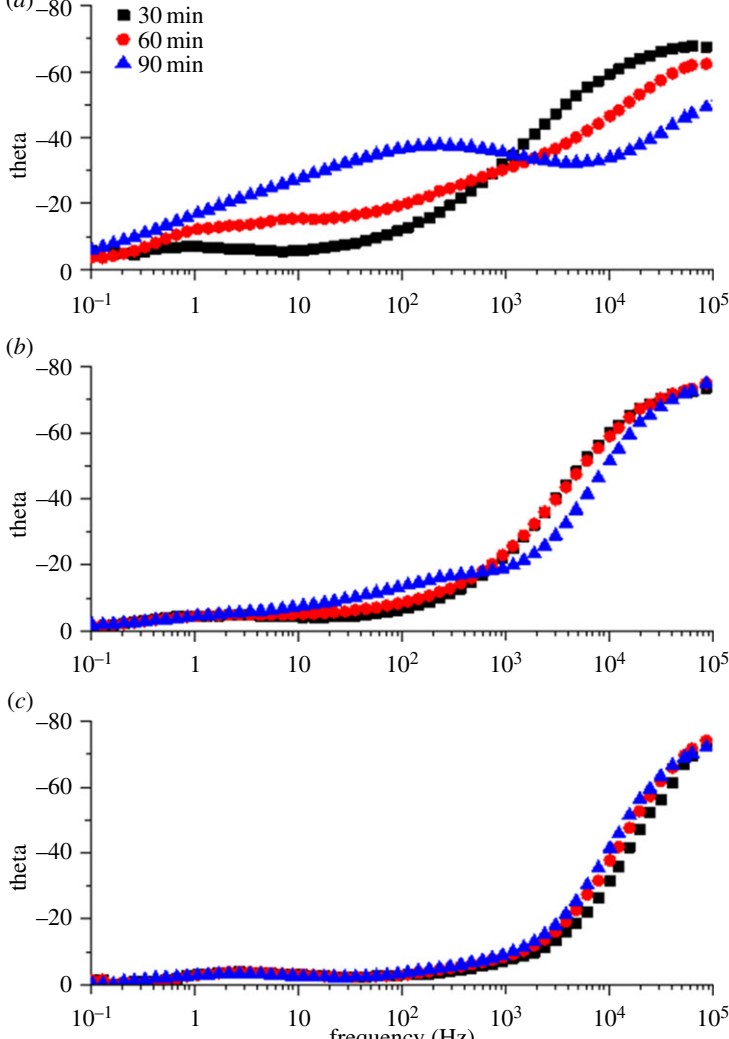

**Figure 7.** Bode plots (phase angle, theta) of (*a*) blank waterborne epoxy coating, (*b*) 0.25 wt% graphene/waterborne epoxy composite coating and (*c*) 0.5 wt% graphene/waterborne epoxy composite coating in $CO_2$-satarated 10 wt% NaCl solution for immersion times of 30–90 min.

indicated the anti-corrosion properties of the composite coating were improved with the addition of graphene content. The composite coating provided effective corrosion protection for the substrate metal during long-term immersion.

Figure 9 shows the equivalent circuit diagram of the EIS profiles, where $R_s$ and $R_c$ stand for the solution resistance and the coating resistance, respectively. $R_{ct}$ is the charge transfer resistance. $Q_c$ is the capacitance of the coating. $Q_{dl}$ is the capacitance of the double layer. The fitting data are shown in table 3. The $R_c$ values for all coating systems decreased and the $Q_c$ values for all coating systems gradually increased with increasing immersion time. This was mainly owing to the gradual penetration of water molecules and corrosive $Cl^-$ with strong polarity and high dielectric constant into the coating. The $R_{ct}$ values decreased with increasing immersion time, because water molecules, $CO_2$ and $Cl^-$ penetrated through the interface between the coating and the base steel, accelerating the corrosion of the steel during immersion.

After 96 h of immersion, the $Q_c$ of the pure epoxy, 0.25 wt% and 0.5 wt% graphene/waterborne epoxy composite coatings were 6.11 μF cm$^{-2}$, $6.34 \times 10^{-4}$ μF cm$^{-2}$ and $1.17 \times 10^{-3}$ μF cm$^{-2}$, respectively. This indicated that graphene enhanced the coatings compactness and prevented the diffusion of corrosive media into the coating. The $R_c$ of the blank epoxy composite decreased from 9.70 kΩ cm$^2$ to 0.455 kΩ cm$^2$, the $R_c$ of the 0.25 wt% graphene/waterborne epoxy composite decreased from 45.2 kΩ cm$^2$ to 20.9 kΩ cm$^2$, while the $R_c$ of the 0.5 wt% composite coating decreased from 227 kΩ cm$^2$ to 117 kΩ cm$^2$ within 96 h of immersion. This indicated that corrosion resistance of the composite

**Figure 8.** Nyquist plots of the (*a*) blank waterborne epoxy coating, (*b*) 0.25 wt% graphene/waterborne epoxy composite coating and (*c*) 0.5 wt% graphene/waterborne epoxy composite coating in $CO_2$-saturated 10 wt% NaCl solution for immersion times of 2–96 h.

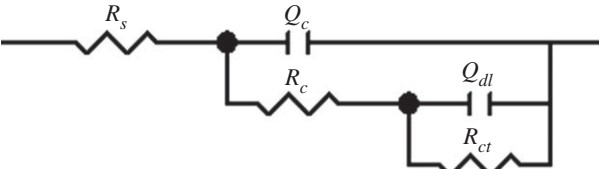

**Figure 9.** Equivalent circuit diagram of the EIS profiles.

coatings gradually increased and remained relatively constant with addition of graphene content. The $R_{ct}$ of the 0.5 wt% graphene/waterborne epoxy composite coating (896 kΩ cm²) was much higher than that of the pure epoxy coating (24.6 kΩ cm²). $R_{ct}$ reflects the difficulty of electron transfer across the interface between electrodes and electrolyte solutions. The addition of graphene increased the resistance of the corrosion reaction of the metal.

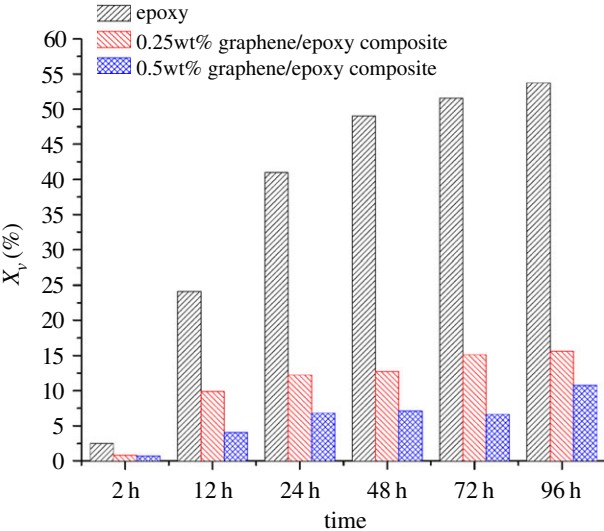

**Figure 10.** Water absorption volume percentage of the different coatings after increasing immersion times.

**Table 3.** Fitting results from the EIS measurements in figure 8.

| | time/h | $R_s$/ ($\Omega$ cm$^2$) | $Q_c$/ ($\mu$F cm$^{-2}$) | $R_c$/ (k$\Omega$ cm$^2$) | $Q_{dl}$/($\mu$F cm$^{-2}$) | $R_{ct}$/ (k$\Omega$· cm$^2$) |
|---|---|---|---|---|---|---|
| epoxy | 2 | 4.23 | 0.647 | 9.70 | 1.03 | 48.5 |
| | 12 | 4.29 | 1.67 | 3.82 | 6.27 | 43.3 |
| | 24 | 3.79 | 3.49 | 1.33 | 14.5 | 39.8 |
| | 48 | 4.86 | 4.97 | 0.961 | 19.3 | 30.3 |
| | 72 | 4.46 | 5.56 | 0.687 | 28.3 | 25.2 |
| | 96 | 3.99 | 6.11 | 0.445 | 40.2 | 24.6 |
| 0.25% wt graphene/epoxy composite | 2 | 3.32 | $3.30 \times 10^{-4}$ | 45.2 | $2.71 \times 10^{-2}$ | 82.04 |
| | 12 | 4.28 | $4.94 \times 10^{-4}$ | 42.7 | $5.63 \times 10^{-2}$ | 69.91 |
| | 24 | 4.37 | $5.47 \times 10^{-4}$ | 39.3 | $3.94 \times 10^{-2}$ | 60.52 |
| | 48 | 4.96 | $5.59 \times 10^{-4}$ | 30.0 | $6.17 \times 10^{-2}$ | 42.43 |
| | 72 | 3.92 | $6.21 \times 10^{-4}$ | 23.5 | 1.64 | 93.87 |
| | 96 | 4.75 | $6.34 \times 10^{-4}$ | 20.9 | 1.85 | 60.01 |
| 0.5% wt graphene/epoxy composite | 2 | 3.98 | $7.53 \times 10^{-4}$ | 227 | $1.26 \times 10^{-2}$ | 1371 |
| | 12 | 3.78 | $8.72 \times 10^{-4}$ | 161 | $1.38 \times 10^{-2}$ | 1290 |
| | 24 | 4.45 | $9.83 \times 10^{-4}$ | 156 | $1.60 \times 10^{-2}$ | 1192 |
| | 48 | 4.60 | $9.97 \times 10^{-4}$ | 142 | $1.71 \times 10^{-2}$ | 1091 |
| | 72 | 4.65 | $9.76 \times 10^{-4}$ | 122 | $1.83 \times 10^{-2}$ | 1166 |
| | 96 | 4.87 | $1.17 \times 10^{-3}$ | 117 | $2.55 \times 10^{-2}$ | 896 |

According to the theory of Brasher and Kingsbury, equation (3.2) can be applied to calculate the water absorption volume percentages of the different coatings [30]:

$$X_v\% = 100 \cdot \frac{\log(C_c(t)/C_c(0))}{\log(80)}, \tag{3.2}$$

where $X_v$ is the water absorption volume percentage of the organic coating, and $C_c(0)$ and $C_c(t)$ are the coating capacitances after immersion for time 0 and time $t$, respectively. In this study, the value of $C_c(0)$ was obtained by the fitting results of EIS measurements for an immersion time of 30 min. The stability of

the water absorption volume percentage can be used to judge whether the performance of the coating is stable. The calculation results are shown in figure 10. The $X_v$ ($t = 96$ h) for the blank epoxy coating, 0.25 wt% and 0.5 wt% graphene/waterborne epoxy composite coating were 53.7%, 15.6% and 10.8%, respectively. The water absorption of epoxy coating decreased significantly with increasing graphene content, and the performance of all the coating systems stabilized after 96 h of immersion. We therefore concluded that the composite coating prevented the diffusion of water into the coating and provided durable and effective anti-corrosion protection for the substrate metal.

# 4. Conclusion

Graphene/waterborne epoxy composite coatings with different mass fractions of graphene were prepared to investigate their corrosion resistance in $CO_2$-satarated NaCl solution. Composite coatings with more uniform surfaces and far fewer defects than blank waterborne epoxy coatings were obtained on 1020 steel. The electrochemical characterizations proved improved corrosion resistance of waterborne epoxy coatings by adding graphene. The 0.25 wt% graphene/epoxy composite coating provided an inhibition efficiency $\eta = 99.96\%$, while further improvement was achieved by increasing the content of graphene additive to 0.5 wt%. The anodic dissolution reaction was retarded by the graphene distributed inside the composite coating, which acted as an effective barrier and suppressed the diffusion of Fe ions. The increase in coating resistance and decrease in water absorption volume percentage of the graphene/waterborne epoxy composite coatings showed that they provided durable and effective protection for the substrate metal in $CO_2$-satarated NaCl solution.

Data accessibility. Data are available in the Dryad Digital Repository: https://doi.org/10.5061/dryad.mkkwh70vk [31].
Authors' contributions. H.X. designed and conducted the experiments. H.H. did the literature search and carried out statistical analyses. H.W. and Y.L. were responsible for data collection. Y.L. did data analysis and language polish.
Competing interests. We declare we have no competing interests.
Funding. This work is supported by the National Key R&D Program of China (grant no. 2017YFF0210406) and the National Natural Science Foundation of China (grant no. 51605368).
Acknowledgements. The authors acknowledge NO.1 Gas Production Plant of Changqing Oilfield Company for the technical and financial support.

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
