## [Reviewer comments · Royal Society Open Science]

Review History

RSOS-191943.R0 (Original submission)

Review form: Reviewer 1

Is the manuscript scientifically sound in its present form?

Yes

Are the interpretations and conclusions justified by the results?

Yes

Is the language acceptable?

Yes

Do you have any ethical concerns with this paper?

No

Have you any concerns about statistical analyses in this paper?

No

Recommendation?

Accept with minor revision (please list in comments)

Comments to the Author(s)

This paper is very interesting and the topic is of highly significance. I recommend a minor revision before its acceptance. The comments and suggestions are as follows.

1. Section 3.2: "The fitting results of polarization curves..." should be modified as "The parameters obtained from the Tafel extrapolation method...". And in Table 2, I suggest to remove the R_p since if you want to get a precise R_p , a special LPR measurement is required.
2. Please mark all of the peaks in Raman and FTIR.
3. The caption of Figure 4 should be: The cross-sectional SEM image...

Review form: Reviewer 2

Is the manuscript scientifically sound in its present form?

Yes

Are the interpretations and conclusions justified by the results?

Yes

Is the language acceptable?

Yes

Do you have any ethical concerns with this paper?

No

Have you any concerns about statistical analyses in this paper?

No

Recommendation?

Major revision is needed (please make suggestions in comments)

Comments to the Author(s)

The authors describe the development and study of an anti-corrosion, CO₂ resistance coating for use in oil and gas industry. The coating consists of a waterborne epoxy with various graphene loadings. The results show that the addition of graphene enhanced the uniformity of the coating and electrochemical testing demonstrated an enhancement in corrosion resistance. The assignment of polarisation curve gradient to diffusion of Fe ions is highly speculative and such effects can be due to a number of processes. The authors should justify this further before publication.

- 1) The discussion of SEM results refers to micropores. Typically, this implies less than 2 nm in pore diameter but here is used to describe larger pores

Decision letter (RSOS-191943.R0)

28-Feb-2020

Dear Dr Xu:

Title: Corrosion Resistance of Graphene/Waterborne Epoxy Composite Coatings in CO₂-saturated NaCl Solution

Manuscript ID: RSOS-191943

Thank you for submitting the above manuscript to Royal Society Open Science. On behalf of the Editors and the Royal Society of Chemistry, I am pleased to inform you that your manuscript will be accepted for publication in Royal Society Open Science subject to minor revision in accordance with the referee suggestions. Please find the reviewers' comments at the end of this email. I apologise that this has taken longer than usual.

The reviewers and handling editors have recommended publication, but also suggest some minor revisions to your manuscript. Therefore, I invite you to respond to the comments and revise your manuscript.

Because the schedule for publication is very tight, it is a condition of publication that you submit the revised version of your manuscript before 08-Mar-2020. Please note that the revision deadline will expire at 00.00am on this date. If you do not think you will be able to meet this date please let me know immediately.

Once again, thank you for submitting your manuscript to Royal Society Open Science. The chemistry content of Royal Society Open Science is published in collaboration with the Royal

Society of Chemistry. I look forward to receiving your revision. If you have any questions at all, please do not hesitate to get in touch.

Best wishes,
Dr Laura Smith
Publishing Editor, Journals

On behalf of the Subject Editor Professor Anthony Stace and the Associate Editor Dr Darren Walsh.

RSC Associate Editor:
Comments to the Author:
(There are no comments.)

RSC Subject Editor:
Comments to the Author:
(There are no comments.)

Reviewer comments to Author:
Reviewer: 1

Comments to the Author(s)
This paper is very interesting and the topic is of highly significance. I recommend a minor revision before its acceptance. The comments and suggestions are as follows.

1. Section 3.2: "The fitting results of polarization curves..." should be modified as "The parameters obtained from the Tafel extrapolation method...". And in Table 2, I suggest to remove the R_p since if you want to get a precise R_p , a special LPR measurement is required.
2. Please mark all of the peaks in Raman and FTIR.
3. The caption of Figure 4 should be: The cross-sectional SEM image...

Reviewer: 2

Comments to the Author(s)
The authors describe the development and study of an anti-corrosion, CO₂ resistance coating for use in oil and gas industry. The coating consists of a waterborne epoxy with various graphene loadings. The results show that the addition of graphene enhanced the uniformity of the coating and electrochemical testing demonstrated an enhancement in corrosion resistance. The assignment of polarisation curve gradient to diffusion of Fe ions is highly speculative and such effects can be due to a number of processes. The authors should justify this further before publication.

- 1) The discussion of SEM results refers to micropores. Typically, this implies less than 2 nm in pore diameter but here is used to describe larger pores

Author's Response to Decision Letter for (RSOS-191943.R0)

See Appendix A.

Decision letter (RSOS-191943.R1)

26-Mar-2020

Dear Dr Xu:

Title: Corrosion Resistance of Graphene/Waterborne Epoxy Composite Coatings in CO₂-saturated NaCl Solution
Manuscript ID: RSOS-191943.R1

Thank you for submitting the above manuscript to Royal Society Open Science. On behalf of the Editors and the Royal Society of Chemistry, I am pleased to inform you that your manuscript will be accepted for publication in Royal Society Open Science subject to minor revision in accordance with the referee suggestions. Please find the reviewers' comments at the end of this email.

The reviewers and handling editors have recommended publication, but also suggest some minor revisions to your manuscript. Therefore, I invite you to respond to the comments and revise your manuscript.

Because the schedule for publication is very tight, it is a condition of publication that you submit the revised version of your manuscript before 04-Apr-2020. Please note that the revision deadline will expire at 00.00am on this date. If you do not think you will be able to meet this date please let me know immediately.

- 1) A text file of the manuscript (tex, txt, rtf, docx or doc), references, tables (including captions) and figure captions. Do not upload a PDF as your "Main Document".
- 2) A separate electronic file of each figure (EPS or print-quality PDF preferred (either format should be produced directly from original creation package), or original software format)
- 3) Included a 100 word media summary of your paper when requested at submission. Please ensure you have entered correct contact details (email, institution and telephone) in your user account
- 4) Included the raw data to support the claims made in your paper. You can either include your data as electronic supplementary material or upload to a repository and include the relevant doi within your manuscript
- 5) All supplementary materials accompanying an accepted article will be treated as in their final form. Note that the Royal Society will neither edit nor typeset supplementary material and it will

be hosted as provided. Please ensure that the supplementary material includes the paper details where possible (authors, article title, journal name).

Kind regards,
Dr Laura Smith
Publishing Editor, Journals

On behalf of the Subject Editor Professor Anthony Stace and the Associate Editor Dr Darren Walsh.

RSC Associate Editor
Comments to the Author:
Please see attached

Reviewer comments to Author:

Author's Response to Decision Letter for (RSOS-191943.R1)

See Appendix B.

Decision letter (RSOS-191943.R2)

03-Apr-2020

Dear Dr Xu:

Title: Corrosion Resistance of Graphene/Waterborne Epoxy Composite Coatings in CO₂-saturated NaCl Solution

Manuscript ID: RSOS-191943.R2

It is a pleasure to accept your manuscript in its current form for publication in Royal Society Open Science. The chemistry content of Royal Society Open Science is published in collaboration with the Royal Society of Chemistry.

On behalf of the Subject Editor Professor Anthony Stace and the Associate Editor Dr Darren Walsh.

RSC Associate Editor
Comments to the Author:
(There are no comments.)

Reviewer(s)' Comments to Author:

Appendix A

Dear Editors and Reviewers,

Thank you very much for taking your time to review this manuscript. I really appreciate all your comments and suggestions. The manuscript (Manuscript ID: RSOS-191943) has been revised by taking all these suggestions into account.

Here are responses to the reviewer comments:

Reviewer #1:

Question 1. Section 3.2: “The fitting results of polarization curves...” should be modified as “The parameters obtained from the Tafel extrapolation method...”. And in Table 2, I suggest to remove the R_p since if you want to get a precise R_p , a special LPR measurement is required.

Answer 1: Thank you for your nice comments on our article. According to your suggestions, the expression of polarization curves in section 3.2 has been modified and the R_p in Table 2 has been removed.

Fixed 1: (Section 3.2, line 3) The parameters obtained from the Tafel extrapolation method are shown in Table 2, in which I_{corr} and E_{corr} stand for the corrosion current density and the corrosion potential respectively. ~~R_p represents the polarization resistance.~~ B_a and B_c represent the slope of the anodic and cathodic polarization respectively.

Question 2. Please mark all of the peaks in Raman and FTIR.

Answer 2: We have added all the functional groups corresponding to the characteristic peaks in section 3.1.

Fixed 2: (Section 3.1, paragraph 3, line 3) The peak near 2800-3000 \$\text{cm}^{-1}\$ was due to the C-H vibration of aliphatic group. The peak in 1244 cm^{-1} and the broad absorption at 3382 cm^{-1} were assigned to C-O-C and -OH respectively. Absorption at 916 \$\text{cm}^{-1}\$ and 827 \$\text{cm}^{-1}\$ were assigned to epoxy groups.

Question 3. The caption of Figure 4 should be: The cross-sectional SEM image...

Answer 3: The caption of Figure 4 has been modified as “The cross-sectional SEM image of 0.5 wt% graphene/waterborne epoxy composite coating on the steel 1020 test surface.”

Fixed 3: (Figure 4, caption)

Figure 4. The cross-sectional SEM image of 0.5 wt% graphene/waterborne epoxy composite coating on the steel 1020 test surface.

Reviewer #2:

Question 1. The assignment of polarization curve gradient to diffusion of Fe ions is highly speculative and such effects can be due to a number of processes. The authors should justify this further before publication.

Answer1: We highly appreciate for your professional review work on our article. As you are concerned, we have not fully discussed the correlation between the slope of polarization curve and the diffusion of iron ions in this work. That is because we discussed this issue in our previous work (H. J. Hu, H. Xu, Y. J. Li, X. Chen, Y. Li, Corrosion NACE. Paper No.13212(2019). Appendix 1). In this work, the line scan by EDS of mild steel coated with 0.5%wt graphene/waterborne epoxy

composite coating and neat waterborne epoxy were measured after immersed in high chlorine CO_2 saturated solution. And we compared the intensity of Fe element inside of the different coatings within $2\ \mu\text{m}$ (figure 1). It can be observed that with the addition of graphene, the intensity of Fe element of 0.5%wt graphene/epoxy composite coating is stronger than that of the pure epoxy coating. We suppose the increase in intensity indicates the deposition of iron ions at the interface of the substrate metal and the composite coating. And The increase of slope of the anodic polarization B_a indicates the transition of anodic reaction. Therefore, we came to a conclusion that the graphene dispersed inside the composite coating retarded the diffusion of iron ions and thus the anodic reaction. The work was cited in the revised manuscript.

Figure 1: The intensity of Fe element inside of the different coatings within $2\ \mu\text{m}$.

Fixed 1:

*(Abstract) This study investigated the corrosion resistance of graphene/waterborne epoxy composite coatings in CO_2 -saturated NaCl solution. The coatings were prepared by dispersing graphene in waterborne epoxy with the addition of carboxymethylcellulose sodium. The structure and composition of the coatings were characterized by SEM, TEM, FTIR and Raman spectroscopies. The corrosion resistance of the composite coatings was investigated by potentiodynamic polarization measurements and electrochemical impedance spectroscopy. Composite coatings with more uniform surfaces and far fewer defects than blank waterborne epoxy coatings were obtained on 1020 steel. The 0.5 wt% graphene/waterborne epoxy composite coating exhibited much lower corrosion rate and provided better water resistance properties and long-term protection than those of the blank epoxy coating in CO_2 -saturated NaCl solution. This was attributed to the distribution of graphene inside the composite coating that retarded the diffusion of iron ions and thus the anodic reaction.

*(Section 3.2, line 16) Table 1 shows that B_a increased with the addition of graphene while B_c did not change significantly. The increase of \$B_a\$ indicates the transition of anodic reaction. With the addition of graphene, the intensity of Fe element of graphene/epoxy composite coating at the interface of the substrate metal and the composite coating is stronger than that of the pure epoxy coating. And the increase in the intensity of Fe element indicates the deposition of iron ions at the interface [28]. Therefore, the graphene distributed inside the composite coating was thought to impede the diffusion of Fe ions which retarded the anodic reaction.

*(References) [28] H. J. Hu, H. Xu, Y. J. Li, X. Chen, Y. Li, Corrosion NACE. Paper No.13212(2019).

Question 2. The discussion of SEM results refers to micropores. Typically, this implies less than 2 nm in pore diameter but here is used to describe larger pores.

Answer 2: Thank you for pointing out the inaccuracy of our choice of words. The expression of “micropores” has been modified as “pores”.

Fixed 2: (Section 3.1, paragraph 6, line 3) The SEM image of the blank waterborne epoxy coating in Figure 5(a) showed a surface topography containing many pores and inhomogeneous defects that is due to the rapid water evaporation during the curing process of waterborne epoxy coatings. These pores and defects will serve as active channels for aggressive species and it is not surprised that the corrosion resistance of waterborne epoxy coatings are significantly lower than those of solvent-based epoxy coatings.

We have corrected them in the revised manuscript by underlining and marking with blue. Thank you again for your consideration of our work!

Yours Sincerely,

Haijun Hu

Xi'an Jiaotong University, Xi'an 710049, China

Appendix 1 :

H. J. Hu, H. Xu, Y. J. Li, X. Chen, Y. Li, Corrosion NACE. Paper No.13212(2019).

Paper No.
13212

Study of CO₂ Corrosion Behavior of Graphene/Epoxy Composite Coating in High Chloride Environment

Haijun Hu^a, Hao Xu^a, Yuejiao Li^a, Xin Cheng^b, Yun Li^a

^a Xi'an Jiaotong University

NO. 28 Xianning West Rd

Xi'an, 710049,

China

^bThe first Natural Gas Plant, PetroChina Changqing Oilfield Company

NO. 10 Changqing South Rd

Jingbian 718500

China

ABSTRACT

Graphene/epoxy composite coatings are prepared through dispersing graphene in waterborne epoxy coatings to investigate the CO₂ corrosion behavior in a high chloride environment. Scanning electron microscopy (SEM) and X-ray photoelectron spectroscopy (XPS) are used to characterize the composite coatings. The corrosion protection performance is studied by electrochemical impedance spectroscopy (EIS) and by measurement of the polarization curve. Results show that the addition of graphene improved the anti-corrosion properties of the composite coatings. The corrosion potential (E_o) of the 0.5 %wt graphene/epoxy composite coating ($E_o = -644.78$ mV) was more positive than that of the epoxy coating ($E_o = -695.78$ mV). The anodic reaction was retarded owing to graphene distributed inside the composite coating that impeded the diffusion of iron ions, while the cathodic reactions were much less affected.

Key words: epoxy coating, graphene, CO₂ corrosion, high chloride

INTRODUCTION

Carbon dioxide flooding as a tertiary recovery method has been widely used all over the world because of its wide application range, high displacement efficiency

©2019 by NACE International.
Requests for permission to publish this manuscript in any form, in part or in whole, must be in writing to NACE International, Publications Division, 1385 Park Two Place, Houston, Texas 77004.
The material presented and the views expressed in this paper are solely those of the author(s) and are not necessarily endorsed by the Association.

Figure 4: The distribution of elements along the depth of mild steel coated with epoxy after electrochemical measurements.

Figure 2 shows the results of the line scan by EDS of mild steel coated with 0.5%wt graphene/epoxy composite coating and the distribution of elements along the depth can be observed in Figure 3. Correspondingly, Figure 4 shows the distribution of elements of mild steel coated with pure epoxy after electrochemical measurements. Take the positions marked in the figure as the origin, and compare the intensity of Fe of different coatings both inside of the base metal and coatings within 2 μm . The results are shown in Figure 5 and Figure 6, respectively. It can be clearly observed that with the addition of graphene, the intensity of Fe of 0.5%wt graphene/epoxy composite coating is two to three times stronger than that of the pure epoxy coating. The increase in intensity indicates the deposition of iron ions at the interface of the substrate metal and the composite coating which demonstrates the barrier effect of the graphene distributed inside the composite coating on the diffusion of iron ions.

©2019 by NACE International
 Requests for permission to publish this manuscript in any form, in part or in whole, must be in writing to
 NACE International, Publications Division, 15835 Park Ten Place, Houston, Texas 77084.
 The material presented and the views expressed in this paper are solely those of the author(s) and are not necessarily endorsed by the Association.

Figure 5: The intensity of Fe inside of the base metal within 2 μm .

Figure 6: The intensity of Fe inside of the different coatings within 2 μm .

©2019 by NACE International
 Requests for permission to publish this manuscript in any form, in part or in whole, must be in writing to
 NACE International, Publications Division, 15835 Park Ten Place, Houston, Texas 77084.
 The material presented and the views expressed in this paper are solely those of the author(s) and are not necessarily endorsed by the Association.

Appendix B

Dear Editors and Reviewers,

Thank you very much for taking your time to review this manuscript. We really appreciate all your comments and suggestions. The manuscript (Manuscript ID: RSOS-191943.R1) has been revised by taking all these suggestions into account.

Here are responses to the comments:

Question 1. The quantities in the tables of data must be converted into standard scientific notation. The authors have used terms such as 6.21E-4 throughout. This is not correct and this example should be presented as 6.21×10^{-4} . Please do this for all quantities throughout the manuscript.

Answer 1: Thank you for pointing out the mistake. According to your suggestions, all the quantities in the tables of data have been converted into standard scientific notation.

Fixed 1:

Table 2. Parameters obtained from polarization curves in Figure 6.

	Bare steel 1020	Blank Waterborne Epoxy	0.25%wt Graphene/Epoxy Composite	0.5%wt Graphene/Epoxy Composite
Corrosion Rate(mm/a)	0.381	7.93×10^{-4}	1.64×10^{-4}	4.99×10^{-5}
I_{corr} (amps/cm²)	3.73×10^{-5}	6.74×10^{-8}	1.39×10^{-8}	4.24×10^{-9}
E_{corr} (Volts)	-0.705	-0.676	-0.639	-0.604
B_a (mV/dec)	41.3	84.7	128	167
B_c (mV/dec)	-357	-329	-289	-345
η	-	99.79%	99.96%	99.98%

Table 3. Fitting results from the EIS measurements in Figure 8.

	Time /h	$R_s/(\Omega \cdot c \text{ m}^2)$	$Q_c/(\mu F \cdot c \text{ m}^2)$	$R_c/(k\Omega \cdot c \text{ m}^2)$	$Q_{dl}/(\mu F \cdot c \text{ m}^2)$	$R_{ct}/(k\Omega \cdot c \text{ m}^2)$
Epoxy	2	4.23	0.647	9.70	1.03	48.5
	12	4.29	1.67	3.82	6.27	43.3
	24	3.79	3.49	1.33	14.5	39.8
	48	4.86	4.97	0.961	19.3	30.3
	72	4.46	5.56	0.687	28.3	25.2
	96	3.99	6.11	0.445	40.2	24.6
0.25%wt	2	3.32	3.30×10^{-4}	45.2	2.71×10^{-2}	82.04

Graphene/Epoxy Composite	12	4.28	4.94×10⁻⁴	42.7	5.63×10⁻²	69.91
	24	4.37	5.47×10⁻⁴	39.3	3.94×10⁻²	60.52
	48	4.96	5.59×10⁻⁴	30.0	6.17×10⁻²	42.43
	72	3.92	6.21×10⁻⁴	23.5	1.64	93.87
	96	4.75	6.34×10⁻⁴	20.9	1.85	60.01
0.5%wt Graphene/Epoxy Composite	2	3.98	7.53×10⁻⁴	227	1.26×10⁻²	1371
	12	3.78	8.72×10⁻⁴	161	1.38×10⁻²	1290
	24	4.45	9.83×10⁻⁴	156	1.60×10⁻²	1192
	48	4.60	9.97×10⁻⁴	142	1.71×10⁻²	1091
	72	4.65	9.76×10⁻⁴	122	1.83×10⁻²	1166
	96	4.87	1.17×10⁻³	117	2.55×10⁻²	896

Thank you again for considering our article for publication in *Royal Society Open Science*.

Yours Sincerely,

Haijun Hu

Xi'an Jiaotong University, Xi'an 710049, China